# Abrasive Sensitivity of Engineering Polymers and a Bio-Composite under Different Abrasive Conditions

**DOI:** 10.3390/ma13225239

**Published:** 2020-11-19

**Authors:** Hasan Muhandes, Ádám Kalácska, László Székely, Róbert Keresztes, Gábor Kalácska

**Affiliations:** 1Faculty of Mechanical Engineering, Institute for Mechanical Engineering Technology, Szent István University, 2100 Gödöllő, Hungary; hasanmuhandes@gmail.com (H.M.); keresztes.robert.zsolt@szie.hu (R.K.); 2Soete Laboratory, Department of Electromechanical, Systems and Metal Engineering, University of Ghent, 9000 Ghent, Belgium; adam.kalacska@ugent.be; 3Institute of Mathematics and Basic Science, Szent István University, 2100 Gödöllő, Hungary; szekely.laszlo@szie.hu

**Keywords:** abrasive wear, engineering plastics, bio-polymer, mechanical properties, regression model, pin-on-plate, slurry

## Abstract

Two different test systems were designed to evaluate the tribological behavior of five engineering plastics (Polyamide—PA grades and Ultra High Molecular Weight Polyethylene—UHMW-PE) and a fully degradable bio-composite (Polylactic Acid—PLA/hemp fibers) targeted to agricultural machinery abrasive conditions. Pin-on-plate tests were performed with different loads, sliding velocity and abrasive particles. The material response was further investigated in a slurry containing abrasive test system with different sliding velocities and distances, abrasive media compositions and impact angles. The abrasive wear, the change of the 3D surface roughness parameters, the friction force and contact temperature evolution were also analyzed as a function of the materials’ mechanical properties ($H,E,\sigma_{y},\sigma_{c},\varepsilon_{B},\sigma_{F},\sigma_{M})$ and the dimensionless numbers derived from them. Using the IBM SPSS 25 software, multiple linear regression models were used to statistically evaluate the measured data and to examine the sensitivity of the material properties and test system characteristics on the tribological behavior. For both test setups, the system and material characteristics influencing the dependent variables (wear, friction, heat generation) and the dimensionless numbers formed from the material properties were ranked using standardized regression coefficients derived from the regression models. The abrasion sensitivity of the tested materials were evaluated taking into account a wide range of influencing parameters.

## 1. Introduction

It is generally accepted that engineering polymers can be advantageously used as a moving machine element due to their favorable tribo-mechanical properties, corrosion resistance and design flexibility [1]. In the design of agricultural machinery, engineering plastics and composites are also of increasing importance in places heavily exposed to abrasive wear. These materials are widely used in harvesters and cultivators, where the abrasion mechanisms can differ significantly. The surface loads that cause the forms of abrasion—i.e., ploughing, wedge formation, micro and macro cutting—[2,3] are extremely different, and the materials must react with different degrees of sensitivity.

Agricultural machine components are affected by typical but markedly different operating conditions that shape abrasive wear. Hence, a wide range of materials are used for these machine elements. Tillage or cultivator implements are characterized by micro-cutting and fatigue acting on their location-specific surfaces [4]; therefore, alloyed martensitic steels are most commonly used. Often, hard alloy coatings are applied to further enhance the wear resistance of these elements [5]. However, specific agricultural machine elements, e.g., parts of a crop harvesting machine, often suffer from abrasive erosion, fatigue and surface cracking, making, in general, the use of polymers more beneficial. The main advantages of polymers are their lubrication free operation (environment protection), low weight (fuel economy and reduction of soil compression by machines) and wear resistance [6]. A persistent problem in commercial grain elevators, often of greater operational importance than kernel damage, is the erosion or wear of equipment at points where grains slide or impinge upon it. Ceramic materials fastened at critical locations resist wear better than alloy steels, but neither cushions the impact. Polyurethanes and specially composed, ultrahigh molecular weight (UHMW) plastics have been developed to provide both wear resistance and reduced impact [7].

However, in order to optimize a given tribological system, it is necessary to know exactly the factors involved (materials, surface parameters, particles, forms of movement, velocity, temperature, intermediates, contaminations, etc.) [8] and to select the most suitable structural materials by modelling.

The tribological literature on polymers and their composites is growing tremendously, in line with the proliferation of industrial applications and the emergence of newer materials. According to a review based on the Web of Science database [9], the number of publications in the field of polymer tribology reached thousands in the second half of the 2000s, accounting for about 25% of the total literature on polymers and composites.

The main mechanisms of polymer wear are adhesion, abrasion and fatigue [10]. Abrasion is caused by hard asperities on the mating surface and hard particles moving on the polymer surfaces. This phenomenon of wear occurs when roughness is a dominant factor during friction processes.

Earlier research on polymer abrasive processes [11,12] described the phenomenon whereby abrasive wear often occurs on surfaces as scratches, holes and pits and other deformed marks. The debris generated by wear is often in the form of fine-cut chips, rather than that generated during machining, albeit in a much finer size [11]. Most models related to abrasive wear have geometric asperity descriptions, so the degree of wear depends heavily on the shape and apex angles of the grinding points moving on the surface. There are two different modes of deformation when an abrasive particle acts upon plastic [11]. The first is plastic grooving, often referred to as ploughing. This occurs when the moving particles or asperities are pushed onto the mating surface and the material is continuously displaced laterally to form grooves and ridges. No material loss on the surface is detected. The second mode is called cutting, because it is similar to micromachining, and all material displaced by the particle is removed as a chip. There is another approach to describing abrasive wear [12]. Experiments have shown that the degree of abrasive wear is proportional to 1/(*σε*), where *σ* is the tensile stress and ε is the corresponding elongation. This connection was found by Lancaster and Ratner in the 1960s [13]. Later in the present article, deeper mathematical correlation analyses highlight the possible connection of abrasive wear features with mechanical property combinations. In papers [14,15], scratch test-based abrasive groove recovery was analyzed and brittleness “*B*”, i.e., less elastic, behavior was calculated with the elongation at break and loss modulus values of the tested polymers.

In engineering practice, the most common technical plastic family comprises the various polyamide 6 (PA6) and polyamide 66 (PA66) variants, as well as the ultrahigh molecular weight polyethylene high density 1000 grade (UHMW-PE HD1000) polymer. There is lot of information available on these materials [9], and a huge amount of research has been done on their abrasion resistance characteristics. Relationships between material properties and specific wear were found, and will be introduced later, but a comprehensive evaluation between combinations of properties and a large number of variables is, at present, absent from the literature.

Rajesh [16] examined two types of PAs. Abrasive wear studies were carried out under a single pass condition by abrading a polymer pin against a waterproof silicon carbide (SiC) abrasive paper with different loads. They found that the CH_2_:CONH ratio had a significant influence on some mechanical properties, e.g., tensile strength, elongation to break, fracture toughness, fracture energy and abrasive wear performance. It was observed that the CH_2_:CONH ratio and various mechanical properties did not show linear relationships in most cases, but the specific wear rate, in function of some mechanical properties (e.g., tensile strength, elongation), showed good correlation.

The same research team presented [17] results about the abrasive wear behavior of numerous PA6-based composites. They applied short glass fiber, polytetrafluoroethylene (PTFE) and metal powders, e.g., copper and bronze, as reinforcing and filling materials. The samples were produced at a lab scale and characterized for their properties such as tensile strength, tensile elongation, flexural strength, hardness and impact strength. In the test system, wear according to hardness, elongation to break (*e*) and ultimate tensile strength (*S*) showed better correlation than that found by Ratner and Lancaster.

Kumara et al. studied the mechanical and abrasive wear behavior [18] of PA6 and glass fiber reinforced (GFR) PA6 composites specimens. The specimens were prepared by injection molding. Four proportions of glass fiber contents were used (0, 10, 20 and 30 wt.%). They performed the test on a pin-on-disc configuration with 320 grit size abrasive paper, at 23 °C and RH 67 ± 10%, 50 m sliding distance at constant sliding velocity (0.5 m/s), with several loads (5, 10, 15 and 20 N). They found that the specific wear rate decreased with increasing sliding distance. This happened because of the polymer formed a transfer layer which filled the space between the abrasive particles, causing a reduced depth of penetration.

Patnaik et al. [19] studied the erosion behavior of solid particles on fiber and particulate-filled polymer composites. Their review focused on problems related to the processes of polymer matrix composites with several aspects, and used the Taguchi method to optimize the process parameters and analyze the wear behavior.

Harsha [20] extended PA6-based composite testing to high performance materials (HPM). Three-body abrasion tests were carried out on more unreinforced thermoplastic HPM polymers by means of a rubber wheel abrasion device. The applied abrasive particle was dry silica sand used as a loose abrasive in the size range between 150 and 250 μm. They applied constant sliding velocity (*v* = 2.4 m/s) of the rubber wheel at different loads (5–20 N). The abrasive wear rates were influenced by the load and type of polymeric materials. On the worn surfaces, it was found that semicrystalline polymers reflected ductile failure mode, whereas amorphous polymers performed brittle failure. Also, an attempt was made to correlate the abrasive wear rates with mechanical properties.

Concerning UHMW-PE HD1000 grade polymer as a widely used, abrasion resistant polymer—unfortunately, due to its low mechanical loadbearing capacity, its engineering applications are limited—the abrasive performance of many PE family members has already been investigated. Tervoort et al. [21] examined the abrasive wear resistance of many PE grades, including UHMW-PE. They found that the effective number of physical cross-links per macromolecular chain influences the abrasive wear resistance.

Nowadays, in addition to the normal use of engineering plastics, more and more attention is being paid to bio-polymers. Research on the wear resistance of bio-polymers and their composites, and the exploration of the peculiarities of the wear processes are appearing independently in more and more settings. From a practical point of view, however, it would be extremely important to compare bio-polymers with already known engineering plastics in a typical test system. Gupta et al. [22] also justified the importance of the developments of bio-composites, focusing on jute fiber reinforced bio-composites.

Mahapatra et al. [23] realized the important effect of deformation and strain on the mechanical applicability of jute fiber reinforced composites, and examined the failure mechanism of the materials. They emphasized that matrix materials must be ductile and not chemically reactive with natural fibers.

Sawpan et al. [24] noted that PLA represents the most common example of a polymer matrix from renewable resources. It has acceptable mechanical behavior for low load applications, but it degrades to carbon dioxide, water and methane after several months to two years, unlike petroleum-based polymers, which need hundreds of years to degrade. Nonetheless, it is considered one of the most important bio-polymers, as noted by Thakur et al. [25]. There are several options for fiber materials that can be used as reinforcement components in bio-composites [26], e.g., flax, hemp, jute and kenaf. The most important disadvantage associated with the use of natural fibers is their hydrophilic nature [27].

The present study explores the extensibility of the abrasion resistance of the aforementioned polyamides and their composites, UHMW-PE and an emerging bio-polymer. In this case, two different test systems were used to evaluate the tribological behavior of five engineering plastics (PA6E (extruded polyamide 6), PA6G (cast polyamide 6), PA6G-ESD (electrostatic dissipative cast polyamide 6 composite), PA66GF30 (extruded polyamide 66 composite reinforced with 30% glass fiber), UHMW-PE HD1000 (ultrahigh molecular weight polyethylene, high density grade “1000”) and a fully degradable bio-composite (PLA/hemp fiber). Pin-on-plate tests were performed with different loads, sliding velocities and abrasive particles to investigate the material behavior. The material response was further investigated in a slurry containing test system (under abrasion conditions) with different sliding velocities and distances, abrasive media compositions and impact angles of the abrasives. The abrasive wear, friction force and contact temperature change of the investigated materials were also analyzed as a function of their mechanical properties ($H,E,\sigma_{y},\sigma_{c},\varepsilon_{B},\sigma_{F},\sigma_{M}$) and the dimensionless numbers formed from them, taking into account different system characteristics.

Concerning the engineering practice, it is clear that agricultural machinery favors the use of the PA6 and PA66 polymer families, as well the UHMW-PE HD1000 grade subjected to abrasive effects during operation. In the literature, there is no comprehensive and global assessment of the abrasion sensitivities of those material families, taking into account the most important mechanical features and the dimensionless numbers that can be formed from them. Furthermore, there are no published results comparing engineering plastics and selected bio-polymers under a wide range of operating condition. Finally, connections between the wear and combined properties of polyamides and UHMW-PE have not yet been presented in the literature.

In this study, using the IBM SPSS 25 software, multiple linear regression models were used to statistically evaluate a large number of measured data, and to examine the sensitivity of material properties and test system characteristics on wear, friction force, heat generation and the change of 3D surface topography. In this way, the abrasion sensitivities of the tested materials were evaluated, taking into account a wide range of influencing parameters.

## 2. Methods and Tested Materials

Two test methods were developed to broadly study the abrasion resistance of the selected engineering polymers and the bio-composite: an abrasive pin-on-plate and a slurry containing system. Using the abrasive pin-on-plate method, micro- and macro- cutting occurred on the polymer surface due to the abrasive particles of standardized commercial clothes, that were originally designed for use as surface cutting/polishing tools. This phenomenon is reportedly dominant—as mentioned in the introduction—with tillage or cultivator elements [4] in a lower speed range (*v* = 0–10 km/h). The slurry containing model can achieve abrasive erosion, which is common for harvester components which are exposed to wet, soil-contaminated products (different plants, e.g., potatoes, rice). Due to abrasive erosion surface fatigue, micro-cracks are often detected in addition to surface groove deformations, wedge formation and cutting [6].

### 2.1. Abrasive Pin-on-Plate Test System

The polymer cylindrical pin specimen (Ø8 mm × 20 mm) with a given normal load (N) slides (m/s) on the abrasive belt moving underneath. Meanwhile, the attached strain gauges measure the abrasion friction force (N), a sensor records the vertical displacement of the clamping head as wear (mm) and the thermocouple measures the temperature change (°C) in the polymer pin at a distance of 8 mm from the contact zone (Figure 1). The data acquisition system contains a Spider 8 A/D converter (Hottinger Baldwin Messtechnik GmbH, Darmstadt, Germany) that passes the digitalized data to computer software. The sampling rate was 5 (1/s) during the measurements. The testing time was set to 10 min on the basis of preliminary measurements; however, not all materials and *pv* sets (Table 1) lasted this long, due to severe wear (*pv*—contact pressure × sliding speed (MPa·ms^−1^) feature of polymeric tribo systems). Those cases are shown in the results later.

In the present research, the following variables were applied:two types of wear interfaces: P60 and P150 standard abrasive clothes;two sliding speeds: 0.031 m/s and 0.056 m/s;three normal loads: 9.81 N, 29.43 N and 49.05 N.

According to these parameters, there were 12 experimental conditions, as listed in Table 1. The extremes of load and speed conditions on both types of abrasive clothing are highlighted. Testing time was 10 min with both speed settings (Table 1), except when the wear was extremely fast. Additionally, after 6–8 mm material loss, the measurements had to be stopped (e.g., PA66GF30 on Figure 5b). The test runs were repeated three times in all 12 cases.

Using this test system, the online wear-, abrasive friction force- and friction temperature change evolution were recorded and specific wear curves were calculated. For the evaluation of the 3D surface parameters of the tested polymers before and after abrasion, a Taylor-Hobson white light microscope (Taylor Hobson Ltd., Leicester, UK) was used, and the 3D parameter values were evaluated using the IBM SPSS 25 software (IBM, Armonk, NY, USA).

### 2.2. Slurry Containing Test System

Figure 2 and Figure 3 show the structure of the applied slurry containing system. An electric motor drives a vertical shaft via a worm gear (1:10) and a clutch. The main holder steel plate is fixed to the shaft. Twelve steel columns holding the polymer specimens are screwed onto the disc from below, arranged on two radii ($r_{1}$ and $r_{2}$) for 6-6 splits of 60 degrees from each other (Figure 3). On the two radii, the columns are offset from each other. The machined-to-size polymer plates (120 mm in length, 20 mm in width and 6 mm in thickness) to be tested were fixed to the two sides of each specimen holder column (Figure 4). The outer radius $r_{1}=280$ mm and the inner radius $r_{2}=200$ mm. The twelve holders were divided into six numbered groups, as shown in Figure 3, i.e., one for each polymer type tested. Each group had an outer and an inner holder, making it possible to test and compare six types of materials with two mean circumferential speeds. The first mean speed was 2.038 m·s^−1^ while the second was 1.456 m·s^−1^.

The entire assembled specimen holder unit was immersed in the abrasive medium and rotated (Figure 2). Based on preliminary experiments, the rotating shaft was placed eccentrically in the pot, providing better mixing of the abrasive medium. During the rotational motion, the particles collided with the polymer plates to be tested. Depending on the location of the polymer plates on the column, “tangential” and “direct” collisions could be distinguished (Figure 3) in the system with different impact mean velocity values (according to $r_{1}$ and $r_{2}$). For the temperature control of the slurry, double pots were applied with cooling water in between; thus, a temperature of 30 °C could be set for the slurry during the tests. The slurry was mixed as a 1:4 volume ratio of water and dry abrasive material.

The polymer samples were tested for five days, with 22 working hours and two hours for wear measurements daily (the duration was determined with preliminary tests to reach the sample’s limit of geometrical loss without abrading the holders). The total abrasive testing time was 110 h for one test series. In some cases, a few samples did not last until the conclusion of the test time due to a combination of the speed, media and position (Figure 3).

The daily 22 working hours can be converted into a straight distance:position 1 & 2: 22 (h) × 60 (min) × 60 (s) × 2.038 (m·s^−1^) = 161,409.6 m;position 3 & 4: 22 (h) × 60 (min) × 60 (s) × 1.456 (m·s^−1^) = 115,315.2 m.

During the daily stopping hours, the wear was measured by the weight loss of the polymer specimens, and their dimensions were measured. For the evaluation of the 3D surface parameters of the tested polymers before and after abrasion, the same method was used as for the pin-on-plate system.

Two abrasive media were selected to make the slurry. One was a gravelly skeletal soil (gravel) and the other was lime coated chernozem (loamy soil). The gravelly skeletal soil is very abrasive, and is one of the most aggressive on machine parts. It is dominated by particles exceeding 2 mm. In some cases, there is significant clay fraction. Lime coated chernozem is the most widely cultivated soil type, owing to its excellent physical characteristics and nutrient management regime. The geotechnical properties of these materials can be found in Table 2.

### 2.3. The Tested Materials

According to the literature, five types of engineering polymers and composites (PA6E, PA6G, PA6G-ESD, PA66GF30, UHMW-PE HD1000) and one kind of bio-composite materials (PLA reinforced by hemp fibers, PLA-HF) were recommended for investigation. The mechanical properties of the tested materials are summarized in Table 3. The engineering polymers and composites were commercial grade, semi-finished plastics, distributed and partly produced by Quattroplast Ltd., Budapest, Hungary. The actual test specimens were machined from the semi-finished rods or plates. The values in Table 3 are according to their certificates. The hardness values in (MPa) used for further combined calculations with other material properties were derived from Shore D dimensionless values. The derived values in MPa give the same order of material hardness as Shore D.

The bio-composite PLA-HF, a fully bio-degradable product, was produced by Boras University (Boras, Sweden). The reinforcing material was cottonized hemp staple fiber (average fiber length 22 mm, average diameter 25–45 μm, density 1.48 g/cm^3^). The polylactic acid fibers were provided by Trevira GmbH (Hattersheim, Germany), with a fiber length of 38 mm and a linear density of 1.7 dtex. This PLA has a density of 1.24 g/cm^3^ and a melting temperature (*Tm*) of 160–170 °C. The final on-woven produced by needle punching composite product contained 40 mass% hemp fibers and 60 mass% PLA.

### 2.4. Evaluation Method

In the online measured pin-on-plate test system, wear as vertical displacement of the specimen holder (mm), calculated specific wear (calculated wear volume under unit load and sliding distance) (mm^3^/N·m), abrasive friction force (N) and friction temperature (°C) evolution were recorded under 12 system conditions (Table 1) according to the sliding distance, *s* (m).

In the slurry containing system, the wear of the real worn surface area (acting with the abrasive slurry to a different extent, depending on the position of the specimen) of the specimen was measured as the decrease of the mass (g) after 22 h of daily operation. The relative wear (%) was calculated as daily weight change percentage compared to the zero day, and the relative wear normalized to km was also defined. The results were compared in terms of the dedicated collision angle and rotational speed (m/s), as shown in Figure 3.

For both test systems, the 3D polymer surface topography was evaluated before and after a given test. The following parameters were measured (ISO 25178 [28]): *Sq* (μm), *Ssk*, *Sku*, *Sp* (μm), *Sv* (μm), *Sz* (μm), *Sa* (μm).

All the measured data were evaluated according to the mechanical properties (Table 3) and in the dimensionless numbers formed from them. A similar method was used for friction analyses of numerous polymers in the adhesive system [29]. The combined dimensionless numbers are:
$\frac{H}{E}$ the ratio between: hardness and elasticity modulus;$\frac{\sigma_{y}E}{\sigma_{M}H}$ the ratio between: combined tensile performance combined bulk-surface stiffness;$\frac{\sigma_{y}H}{\sigma_{M}E}$ the ratio between: combined surface strength/combined strength-stiffness;$\frac{\sigma_{F}\sigma_{y}}{\sigma_{M}H}$ the ratio between: combined tensile-flexural strength/combined strength-hardness;$\frac{E}{\sigma_{C}}$ the ratio between: elasticity modulus/compression strength;$\frac{\sigma_{F}}{\sigma_{C}}$ the ratio between: flexural strength/compression strength;$\frac{H\varepsilon_{B}}{\sigma_{y}}$ the ratio between: combined Hardness-strain capability/Yield strength;$\frac{\sigma_{C}\varepsilon_{B}}{\sigma_{M}}$ the ratio between: combined compression-strain capability/tensile strength;$\frac{\sigma_{y}}{\sigma_{C}\varepsilon_{B}}$ the ratio between: Yield strength/combined compression-strain capability;$\frac{\sigma_{F}H}{\sigma_{M}E}$ the ratio between: combined Flexural performance/combined bulk-surface stiffness.

For the statistical analyses, multiple linear regression models were developed using the IBM SPSS 25 software. To examine how a dependent variable depended on several independent (or explanatory) variables which were all measured at different scales, the main tool was multiple regression. In this paper, it was verified that the functions of several variables which describe this dependence were approximately linear, that is, for a dependent variable:(1)$$Y=a_{0}+a_{1}X_{1}+a_{2}X_{2}+\cdots +a_{n}X_{n},$$
where n is the number of independent variables and $X_{1},X_{2},\ldots ,X_{n}$ are the independent variables.

The method of least squares is the most common way to fit such a model to the measured data. First, an F test was always carried out to see whether the corresponding model was relevant. If the p-value was less than 0.05, that is, it was significant, then the model was relevant. Usually, the $R^{2}$ value measures the goodness-of-fit of such a linear model, i.e., it shows how much of the variance of the dependent variable may be explained by the independent variables. If p<0.05 holds for coefficient $a_{k}$
(k∈{0,1,…,n}), then it is statistically different from 0, and thus, the corresponding independent variable $X_{k}$ plays a role in describing the dependence of Y. In the discussed models, only the explanatory variables are included for which their associated coefficient turned out to be significant. To see which independent variable had the greatest effect on Y, one must consider the absolute value of the standardized (or beta) coefficients of the significant independent variables; the higher this value is, the greater the effect. Among the possible methods of entering variables into a linear regression model, the stepwise method was used. This means that at each step of the model building algorithm, among the possible, significant independent variables, the one which caused the highest change in $R^{2}$ was entered. The algorithm ended when there were no new independent variables to enter. In some cases of the below presented models, not all significant variables were included; e.g., those causing very low change in $R^{2}$ were neglected. It is important to mention that one of the assumptions of the applicability of multiple linear regression is that the independent variables are not collinear. Since many of the parameters of the studied materials have high correlation coefficients with another parameter, not all of them were used in the models at the same time.

## 3. Results and Evaluation

### 3.1. Pin-on-Plate Sytem

Concerning the online registered large database, the extremes of test conditions (maximum and minimum loads and speeds, numbered 1, 6, 7, 12 in Table 1) on two types of abrasive clothes are shown in the following diagrams. Figure 5 shows the P60 and Figure 6 the P150 results: wear based upon sliding distance with a linear approach.

Figure 5a refers to the first test condition (Table 1) having the lowest speed and load (lowest pv) on the P60 surface, where all the materials could slide the same distance. It shows the online measured wear according to the the sliding distance. Similarly, Figure 6a plots the wear on the P150 surface under the same pv. As shown in Figure 5a, the cast polyamide 6 performed the best, and PA66GF30 the worst. At the applied top speed and load, which was the applied highest *pv* (Table 1, test condition 6), different sliding distances can be seen (Figure 5b) according to the limit of fast wear and the specimen dimensions. In that case, PA6G also offered the best wear resistance. Using P150 abrasive clothes, the UHMW-PE HD1000 offered the best wear resistance under high *pv* conditions (Figure 6b).

The measured online wear points of the twelve system conditions (Table 1) were approached by linear fit. The slope values “a” of the linear regression (y=ax+b) are summarized in Table 4, which makes it possible to compare the wear speeds.

The calculated specific wear curves for the same cases and materials are introduced in Figure 7 and Figure 8, focusing on the running-in phases of the tests, where the starting sensitivity and differences can be expressed. The first half meter of sliding is critical from the point of specific wear that dictates the slopes (Table 4) and positions of the lines (Figure 5 and Figure 6). It is clear that the PLA-HF bio-composite was not worse than the average of the tested engineering polymers in this pin-on-plate system. On both P60 and P150 abrasive surfaces, it can be ranked as 4–5th in terms of wear resistance, i.e., similar to UHMW-PE or PA6 ESD, according to the *pv*.

In parallel with the wear curves, the arising friction force and temperature change in the plastic pin were also recorded. A large amount of data was evaluated using the developed statistical models. A typical plot can be seen in Appendix A (Figure A1).

The graphs in Figure 9 show that the wear values taken from the end region of the tests plotted against the dimensionless numbers of the materials can offer connections typically within the polyamide family, regardless of being a natural or composite one.

Taking all the results into account, the following findings can be stated:
There are proportional relationships between the wear values of the polyamides (PA66GF30, PA6G ESD, PA6G and PA6E) and $\frac{\sigma_{y}E}{\sigma_{M}H},$ as well as $\frac{\sigma_{y}}{\sigma_{C}\varepsilon_{B}}$. By the increasing dimensionless number values, a higher degree of wear was measured. A similar trend was observed with the polyamides and UHMW-PE HD1000 in case of $\frac{E}{\sigma_{C}}$.There were inverse relationships between values $\frac{H}{E},\frac{\sigma_{y}H}{\sigma_{M}E}$ and the wear of polyamides (PA66GF30, PA6G ESD, PA6G and PA6E). By increasing these dimensionless number values, a lower degree of wear was measured.$\frac{\sigma_{F}\sigma_{y}}{\sigma_{M}H}$ presents a proportional relationship with wear, whereby five materials followed the trend but UHMW-PE HD100 did not.These findings are in accordance with previously published data [16,17,18] on the main mechanical properties; however, the above analyzed combined dimensionless numbers were not studied. No connection between wear and the combined properties of polyamides and UHMW-PE has been published to date.

Multiple linear regression models analyzed the sensitivity of material properties and test system characteristics on wear, friction force and heat generation, separately on P60 and P150 abrasive surfaces. In the test systems, the sliding distance “*s*”, the load “*F_N_*”, and the sliding velocity “*v*”, were considered as independent variables, as were the material properties and the indicators derived from them.

#### 3.1.1. Wear, Using P60 Abrasive

The best possible fitting model was:(2)$$\mathrm{wear}=a_{0}+a_{1}s+a_{2}F_{N}+a_{3}\cdot \frac{\sigma_{y}}{\sigma_{c}\varepsilon_{B}}+a_{4}H$$

The F-value of the model was $9.619\times 10^{4}$ and p<0.001; thus, this model is relevant. The coefficients of the model are summarized in Table 5. For this model, the goodness-of-fit was $R^{2}=0.831.$ Time of the experiment had the greatest effect on the wear of the test sample, while among the material parameters, $\frac{\sigma_{y}}{\sigma_{c}\varepsilon_{B}}$ had some effect.

#### 3.1.2. Wear, Using P150 Abrasive

The best possible fitting model was:(3)$$\mathrm{wear}=a_{0}+a_{1}s+a_{2}F_{N}+a_{3}v+a_{4}\cdot \frac{\sigma_{y}}{\sigma_{c}\varepsilon_{B}}$$

The F-value of the model was $2.112\times 10^{5}$ and p<0.001; thus, this model is relevant. The coefficients of the model are summarized in Table 6.

For this model, the goodness-of-fit was $R^{2}=0.890.$ Time of the experiment had the greatest effect on the wear of the test body, while among the material parameters, $\frac{\sigma_{y}}{\sigma_{c}\varepsilon_{B}}$ had some effect.

#### 3.1.3. Abrasive Friction Force, P60

Without the table of coefficients of the regression model, in the case of the P60 abrasive, the best possible fitting model was:(4)$$F_{f}=-1.427+10.351F_{N}-0.022\varepsilon_{B}+0.243\cdot \frac{\sigma_{c}}{\sigma_{y}\varepsilon_{B}}$$

The F-value of the model was $3.370\times 10^{6}$ and p<0.001; thus, the model is relevant. For this model, the goodness-of-fit was $R^{2}=0.99.$ Still load had the greatest effect on the resultant friction force, while among the material parameters, $\frac{\sigma_{F}\sigma_{y}}{\sigma_{M}H}$ and H had minor effects.

#### 3.1.4. Abrasive Friction Force, P150

With P150 abrasive, the best fitting model was:(5)$$F_{f}=-4.951-0.084s+9.051F_{N}+1.424H-1.133\cdot \frac{\sigma_{F}\sigma_{y}}{\sigma_{M}H}$$

The F-value of the model was $5.207\times 10^{4}$ and p<0.001; thus, the model is relevant. For this model, the goodness-of-fit was $R^{2}=0.983.$ Still load had the highest effect on the resultant friction force, while among the material parameters, $\frac{\sigma_{F}\sigma_{y}}{\sigma_{M}H}$ and H had some effect.

#### 3.1.5. Friction Heat Change, P60

Concerning friction heat evolution on the P60 abrasive clothes measured close to contact inside the polymer pin, the best fitting model was (not showing the detailed coefficients table):(6)$$\Delta T=17.502+0.349s+2.618F_{N}+107.796v+0.229H-0.163\sigma_{F}-1205.488\cdot \frac{\sigma_{F}H}{\sigma_{M}E}-0.108\cdot \frac{H\varepsilon_{B}}{\sigma_{y}}$$

The F-value of the model was 3654 and p<0.001; thus, the model is relevant. For this model, the goodness-of-fit was $R^{2}=0.728.$
$\sigma_{F}$ had a very high effect; furthermore, load, distance and $\frac{\sigma_{F}H}{\sigma_{M}E}$ had a light effect on the temperature increase during sliding.

#### 3.1.6. Friction Heat Change, P150

With P150 abrasive, the heat generation is approximated as:(7)$$\Delta T=8.787+0.158s+1.535F_{N}+70.251v+0.011\varepsilon_{B}-1.009H-723.359\cdot \frac{\sigma_{F}H}{\sigma_{M}E}$$

The F-value of the model was $6.797\times 10^{4}$ and p<0.001; thus the model is relevant. For this model, the goodness-of-fit was $R^{2}=0.794.$ Load, H and $\frac{\sigma_{F}H}{\sigma_{M}E}$ had the highest effect on the temperature change measured.

Concerning the change of the polymer surfaces, white light 3D microscopy was used. Table 7 gives a summary of the original 3D surface characteristics, and Figure 10 shows scaled.

In Figure 10, it can clearly be seen that the tough HD1000 suffered essential deformation, with new hills on the surfaces, while the less tough PA66GF30 presented new and deep, wide grooves, i.e. wear. The microscopic pictures are in good agreement with the measured wear (Figure 5 and Figure 6). The change of the surface parameters was systematically evaluated for the selected test conditions (extremes cases, shown in Table 1), and the data were analyzed by multiparameter regression models. The change of the parameters in the system condition No. 6 can be seen in Table 8.

Once more, multiple linear regression models were constructed where the dependent variables were the 3D surface parameters and the independent ones were the sliding distance, the load, the calculated *pv* and the sliding velocity, as well as the material properties and the derived indicators. Concerning the results, the wear interface P60 goodness-of-fit varied between 0.25 and 0.47, and usually, the only explanatory variable was the sliding distance, except with *Ssk*, where instead of s, $\sigma_{F}$ appeared. Also, it should be mentioned that for *Sku*, there was not any suitable linear model for the process. This was mainly due to the high deviation of this parameter in the examined materials. The P60 abrasive surface acted as a real cutting one at almost a mm scale, where the measure of the standard parameters may start to fail. Regarding the much smoother wear interface P150 for the 3D parameters *Ssk*, *Sku* and *Sp*, the goodness-of-fit was low (0.2–0.4), but for the others, it ranged from mid to high; the obtained models are presented below in detail.

#### 3.1.7. *Sq*, P150

For *Sq*, the best possible fitting model was:(8)Sq=6.057+0.052s−0.00014E

The F-value of the model was 15.989 and p=0.001; thus, the model is relevant. For this model, the goodness-of-fit was $R^{2}=0.804,$ and E had the highest effect on *Sq*.

#### 3.1.8. *Sv*, P150

For *Sv*, the best possible fitting model was:(9)Sv=12.790+0.036s−0.0003302E

The F-value of the model was 54.157 and p<0.001; thus, the model is relevant. For this model, the goodness-of-fit was $R^{2}=0.861,$ and E had a high effect on *Sv*.

#### 3.1.9. *Sz*, P150

For *Sz*, the best possible fitting model was:(10)$$Sz=2.908+0.142s+0.031\cdot \frac{\sigma_{F}}{\sigma_{c}}$$

The F-value of the model was 12.960 and p<0.001; thus, the model is relevant. For this model, the goodness-of-fit was $R^{2}=0.552,$ and $\frac{\sigma_{F}}{\sigma_{c}}$ had the highest effect on *Sz*.

#### 3.1.10. *Sa*, P150

For *Sa*, the best possible fitting model was:(11)Sa=5.018+0.032s−0.0001164E

The F-value of the model was 38.520 and p<0.001; thus, the model is relevant. For this model, the goodness-of-fit was $R^{2}=0.786,$ and E had the highest effect on *Sa*.

According to the presented results, one can see that the elasticity modulus and flexural and compressive strength play roles in the change of surface parameters under a much smoother (P150) abrasive condition compared to P60.

These findings are in accordance with the effects of contact loads on the microgeometry. The microgeometry of the moving polymer surfaces under a given normal load mainly suffer shear, bending and compressive effects, that can ultimately lead to the appearance of deformation, wedge formation and microcutting [2,3]. According to the shear, bending and compressive loads, the models proved the roles of $\sigma_{F}$ and $\sigma_{c}$, as well as of the Young’s modulus. Other tensile values did not play significant roles.

### 3.2. Abrasive Sesitivity of the Tested Materials in a Pin-on-Plate Sytem

As the final statistical evaluation of the pin-on-plate test results, the novelty of the research results are summarized:Introducing the abrasive sensitivity of the tested materials based on the standardized coefficients of multiple regression models.Ranking the abrasive sensitivity of the materials according to the independent variables of the test systems.

Consider a dependent variable (e.g., wear, etc.) and the material properties and indicators which have a significant effect on it. The abrasive sensitivity is the extent to which the independent variables affect it, which is related to the standardized regression coefficients. Therefore, the higher the absolute value of the corresponding standardized regression coefficient, the higher the abrasive sensitivity of the dependent variable (wear, friction, temperature, surface 3D properties) with respect to the independent variable.

Based on the considered linear models, Table 9 shows the abrasive sensitivity for wear, abrasive friction force, heat generation and 3D parameters. The factors are in increasing order of effect.

### 3.3. Slurry Containing System

In the slurry containing system, as described in detail in Section 2.2., the material samples moving in the slurry medium at two circumferential speeds and at two angles of impact suffered abrasive erosion on the surface. As previous tests proved, with many structural materials and coatings, micro-cutting, polishing and pitting (surface fatigue due to particle impact) can occur on the surfaces according to the features of the applied slurry containing system [30]. In the following cases, where gravelly skeletal and loamy soils were used as abrasive media, the micro-cutting caused continuous material loss, leaving grooves on the partly polished surfaces. Our primary assessment was based on the weight loss of the samples plotted against the wearing distance in the slurry. The effects of speed and collision angle are also analyzed by the regression models. The change of 3D surface parameters were evaluated similarly to pin-on-plate systems. Figure 11 shows the relative wear (%) of the different materials (calculated as an actual (daily cumulative) weight loss in percentage, compared to the zero day weight) according to the four different positions (Figure 3) in the case of gravelly skeletal soil, which is considered the most abrasive soil. Figure 11a,b shows longer daily distances for the $r_{1}$ radius positions 1, 2 (Figure 3) due to the higher speed. Figure 12 presents the same wear evaluation in the case of the loamy soil-based slurry. In both cases, it was clear that PLA-HF composites were more sensitive for the slurry containing conditions, as was found for pin-on-plate abrasive dry cutting. On the dry pin-on-plate system, the bio-degradable composite demonstrated average wear resistance compared to the engineering plastics; meanwhile, in the gravel slurry system, the PLA-HF suffered much severe material loss and was not comparable with engineering polymers. In this type of plot, no essential differences were detected among the engineering polymers with the gravel slurry.

In the case of a good quality loamy soil for growing cereals, the abrasive erosion comprised less contrast between PLA-HF and engineering polymers. Still, the bio-composite performed best in terms of wear, but not so differently as with gravels. Also, a difference among the engineering polymers can be seen with the loamy soil. While the PA6E performed best with regard to wear resistance in the case of gravel, with the loamy slurry media, PA6G was the best. This was due to different factors in the tribo systems, which are evaluated by multiple regression models.

The speed of wear between the days varied; thus, daily evaluations were also performed (calculated as daily weight change in percentage compared to the previous day’s weight). Figure 13 shows an example of the position 1 results with both two types of slurry.

It is clear that the PLA-HF performed a running-in like effect over the first two days, leading to the fast wear of the coating layer of the composites. Later, the fibers of the bulk suffered a lower daily wear. Concerning the engineering polymers, the virgin PA6E, PA6G and UHMW-PE HD1000 showed less fluctuation in daily wear than the composite PA66GF30 and PA6G ESD. This observation is in accordance with previous studies on composite materials in abrasive systems [16,17], since the reinforcements—beside the generally positive effect on mechanical properties—can also cause uneven internal stresses that influence the wear behavior.

Multiple linear regression models were applied to statistically investigate the sensitivity of the material properties on wear for both gravel and loamy soil. In the test systems, the duration of the test t (measured in days), the tangential velocity v of the test specimen and the contact angle were considered as independent variables, as were the material properties and the indicators formed from them. It should be mentioned that the contact angle was treated as a dummy variable, that is, it was defined as 1 if its position was direct, and as 0 if its position was tangential (Figure 3). Therefore, the notation ca as “contact angle” was used. The following models were obtained for the two types of soil.

#### 3.3.1. Wear, Gravel

Without presenting all the details of the fitted model, the best fitting one was:(12)$$\mathrm{wear}=0.018+0.007t-0.010v-0.009ca+0.0002434\cdot \frac{\sigma_{y}}{\sigma_{c}\varepsilon_{B}}.$$

The F-value of the model was 131.897 and p<0.001; thus, the model is relevant. For this model, the goodness-of-fit was $R^{2}=0.791.$
$\frac{\sigma_{y}}{\sigma_{c}\varepsilon_{B}}$ and the time of the experiment had the greatest effect on the wear.

#### 3.3.2. Wear, Loamy Soil

The best fitting model was:(13)$$\mathrm{wear}=0.011+0.003t-0.004v-0.004\mathrm{ca}-0.0000260\varepsilon_{B}-0.0000255\sigma_{F}+0.0000033\cdot \frac{\sigma_{y}E}{\sigma_{M}H}.$$

The F-value of the model was 145.729 and p<0.001; thus, the model is relevant. For this model, the goodness-of-fit was $R^{2}=0.865.$ Among the material properties, $\frac{\sigma_{y}E}{\sigma_{M}H}$ had the highest effect on wear, albeit a mediocre one.

The evaluation of the 3D surface topographic changes was similar to that already presented for the pin-on-plate system. For both slurry abrasive media, the state before and after measurement was compared according to the four positions (Figure 3) for each polymer. Figure 14 shows, by way of example, the microscopic visualization of the polymer surfaces in the initial and final states with the gravel medium at position 1. The measured values of surface parameters and their changes are summarized in Table 10 and Table 11. The complete surface topographic database of slurry measurements was analyzed in detail with the regression modeling already described.

Comparing Figure 14 and Figure 15 and the characteristic parameters, it is clear that by the end of the slurry abrasive erosion, the original machining marks of the surfaces had completely disappeared due to the swirling slurry. In all cases, the result was a polished surface with different degrees of moderate hills and valleys partly interrupted by microgrooves.

To examine the dependence of the 3D surface parameters on the duration of the test, the tangential velocity of the test specimen, the contact angle and the material properties, multiple linear regression models were constructed. Unfortunately, for both types of soil, due to the high deviation of the 3D parameters of the examined materials for most of the models, the goodness-of-fit varied between 0.12 and 0.4, and usually, the only explanatory variable was the duration of the experiment. For gravel, the only model worthy of mention was *Ssk*, where although $R^{2}=0.449,$
E and $\frac{\sigma_{y}E}{\sigma_{M}H}$ appeared as explanatory variables. For loamy soil, the two most interesting models were fitted for *Sq* and *Sa*; they are presented in detail below.

#### 3.3.3. *Sq*, Loamy Soil

For *Sq*, the best possible fitting model was:(14)Sq=5.320−0.674t−0.0000543E
The F-value of the model was 25.195 and p<0.001; thus, the model is relevant. For this model, the goodness-of-fit was $R^{2}=0.604,$ and E had a mediocre effect on *Sq*.

#### 3.3.4. *Sa*, Loamy Soil

For *Sa* the best possible fitting model was:(15)Sa=4.473−0.581t−0.0000504E

The F-value of the model was 28.955 and p<0.001; thus, the model is relevant. For this model, the goodness-of-fit was $R^{2}=0.637,$ and E had a mediocre effect on *Sa*.

Similarly to the abrasive sensitivity of the pin-on-plate system, the same concept can be introduced for the slurry systems as well. The results, which are again based on the (unpresented) standardized regression coefficients of the corresponding linear models, are summarized in Table 12.

## 4. Conclusions

Abrasive pin-on-plate and slurry containing systems were applied to study of the abrasive behavior of five engineering polymers and a bio-degradable composite. The two test systems represent two abrasive loads, where the cutting (microcutting) and fatigue effects on the surfaces are different. Applying both approaches to evaluate abrasion sensitivity is more comprehensive. The measured friction, wear and heat generation and 3D surface change were analyzed in conventional ways (graphs, microscopic photos), as well as by means of multiple linear regression models, developed using IBM SPSS 25 software. In the test systems, the sliding distance, load, sliding velocity, material properties and indicators formed from them and the contact hit angle were considered as independent variables.

Concerning the tested materials, the “sensitivity to abrasion” was introduced, based on the multiple linear regression models, taking the standardized beta regression coefficients into account. The coefficients showed which independent variables had the most significant effect on the measured tribo data concerning both test systems.

In the layout of the abrasive pin-on-plate systems (on P60 and P150 particles), where the deformation, micro-cutting and cutting effects were dominant:
PA6G offered the best abrasive resistance, as PA66GF30 the worst. The bio-polymer (PLA-HF) was average among the engineering polymers.There were proportional relationships between the wear values of the polyamides (PA66GF30, PA6G ESD, PA6G and PA6E) and $\frac{\sigma_{y}E}{\sigma_{M}H},$ as well as $\frac{\sigma_{y}}{\sigma_{C}\varepsilon_{B}}$. By the increasing dimensionless number values, a higher degree of wear was measured. A similar trend was observed for the polyamides and UHMW-PE HD1000 in the case of $\frac{E}{\sigma_{C}}$.There were reciprocal relationships between values $\frac{H}{E},\frac{\sigma_{F}H}{\sigma_{M}E}$ and the wear of polyamides (PA66GF30, PA6G ESD, PA6G and PA6E). By increasing these dimensionless number values, a lower degree of wear was measured.$\frac{\sigma_{F}\sigma_{y}}{\sigma_{M}H}$ offered a proportional relationship with wear whereby five materials followed the trend-line, while UHMW-PE HD100 did not.Despite the different abrasive surfaces, similar material characteristics affected the ΔT. $\sigma_{F}$ and $\frac{\sigma_{F}H}{\sigma_{M}E}$ were dominant parameters, which reflects the accumulated work of the microgeometrical deformation partly converted to heat.Beyond the published results in the literature (in which mainly tensile and hardness properties are connected to abrasion behavior), the multiple regression models proved their high sensitivity and relationship with $\sigma_{F},\;\sigma_{C}$ alone and in derived dimensionless forms. The change of surface 3D parameters correlated mainly with *E* and $\sigma_{F},\;\sigma_{C}$.

In the layout of the slurry containing systems (gravel and loamy soil-based slurry), where abrasive erosion (beside the cutting effect, the particle impact with different angles and speeds) plays a role:
PLA-HF showed the worst abrasive resistance, while the engineering polymers offered similar wear trends; PAE and PA6G were the best.Concerning the wear speed on a daily basis, the PLA-HF showed a running-in like effect during the first period of testing, which comprised fast wear of the coating layer. The virgin polymers showed less fluctuation in daily wear than the composite PA66GF30 and PA6G ESD. This may have been caused by the uneven internal stress distribution due to the reinforcements.The sensitivity analyses in the abrasive erosion systems enhanced the primary role of the tensile features, e.g., $\frac{\sigma_{y}E}{\sigma_{M}H,\;}$, E, and $\varepsilon_{B}$ while the compressive and flexural properties, unlike with the pin-on-plate, played lesser roles.

Sensitivity analyses showed that the varying abrasive load conditions presented different material characteristic groups (e.g., tensile, compressive, flexural). Some parameters appeared in more system conditions:
Wear P60, P150 and gravel: $\frac{\sigma_{y}}{\sigma_{C}\varepsilon_{B}}$ is important among the material characteristics. The appearance of the compressive strength is in accordance with the mode of the complex load of the micro-geometries.In slurry containing and pin-on-plate, *E* is important for changing the 3D surface parameters. This reflects on the role of deformation capability of surface micro-geometry.Comparing the two systems, beside the sliding distance and load, the following material properties and dimensionless numbers occurred as influencing factors: $\frac{\sigma_{y}}{\sigma_{c}\varepsilon_{B}}$ had an effect on the wear rate in both systems, while for the micro-geometric change of pin-on-plate, in the case of P150, and the slurry in the case of loamy soil, similar sensitivities to the *E* modulus of the materials were revealed.

## Figures and Tables

**Figure 1 materials-13-05239-f001:**
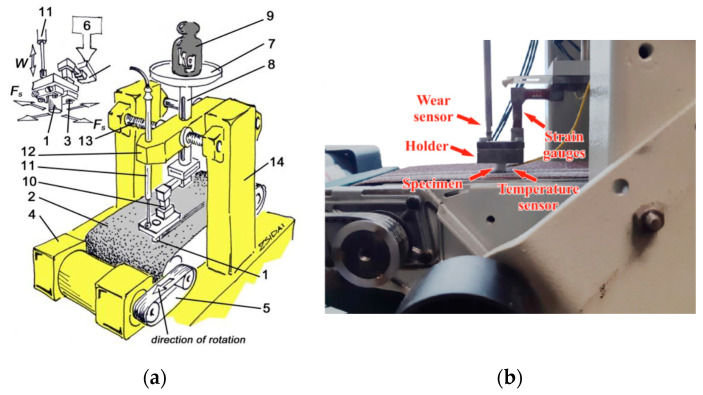
Schematic figure (**a**) and a photo (**b**) about the abrasive pin-on-plate test system. (1) polymer specimen; (2) abrasive cloth; (3) nuts and clamp; (4) electrical motor; (5) twin roll driving system; (6) manual loading system; (7) plate; (8) vertical column; (9) dead loading weights; (10) strain gauges; (11) linear gauge (for vertical displacement as a result of wear); (12) console head unit; (13) spindle for cross movement (it wasn’t used for present tests).

**Figure 2 materials-13-05239-f002:**
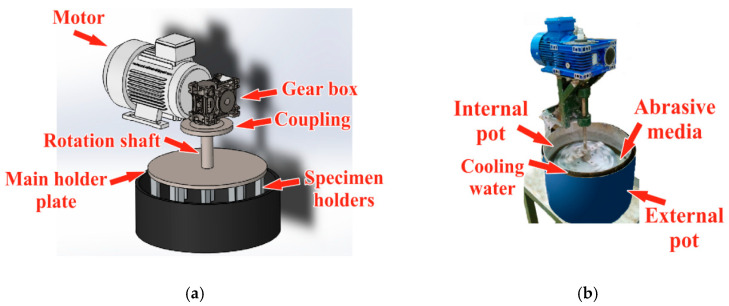
Assembly of the slurry containing test system. (**a**) Structure; (**b**) In operation.

**Figure 3 materials-13-05239-f003:**
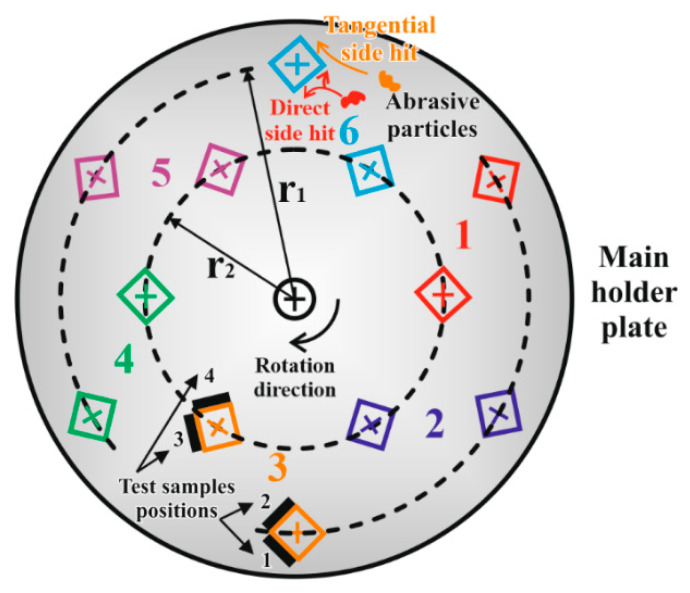
Layout of the 12 holder columns on the main holder plate. Tangential and direct collision, as well as the numbered polymer sample positions (1–4), are indicated.

**Figure 4 materials-13-05239-f004:**
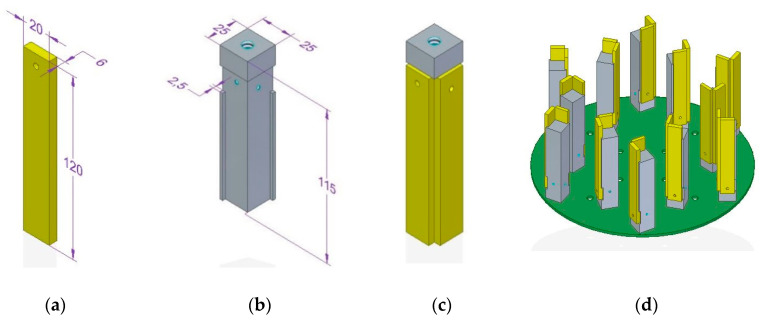
Assembly line of the specimen system: polymer sample, steel holder column, samples placed on the column, and finally, the holder unit in an upside-down position: (**a**) test specimen; (**b**) specimen holder; (**c**) two specimen placed on holder; (**d**) layout on the disc, upside down drawing.

**Figure 5 materials-13-05239-f005:**
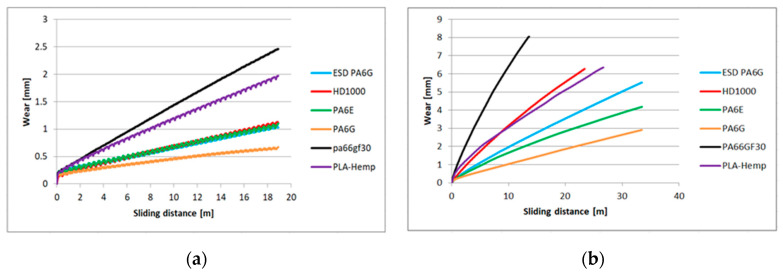
The relationship between the wear and sliding distance for several polymer types on P60 abrasive clothing: (**a**) lowest applied *pv*, (**b**) highest applied *pv*.

**Figure 6 materials-13-05239-f006:**
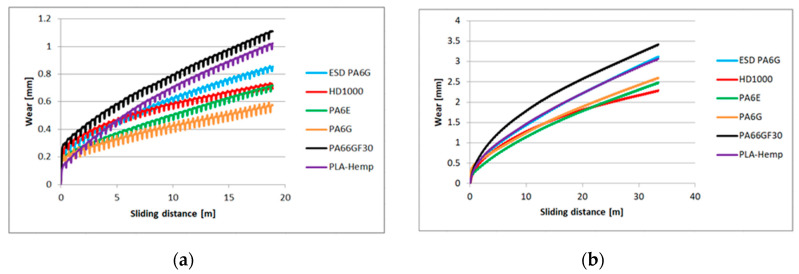
The relation between the wear and sliding distance for several polymer types on P150 abrasive clothing: (**a**) lowest applied *pv*, (**b**) highest applied *pv*.

**Figure 7 materials-13-05239-f007:**
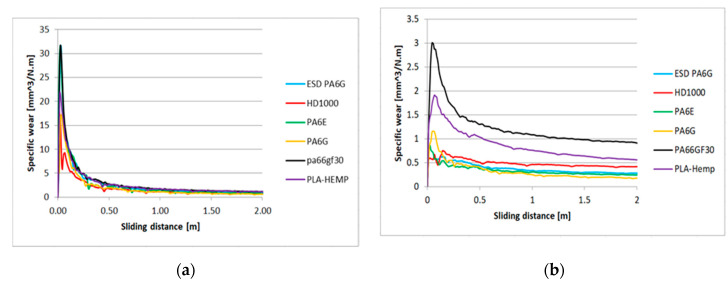
The relation between the specific wear and sliding distance for several polymers types on P60 abrasive clothing” (**a**) lowest applied *pv*, (**b**) highest applied *pv* (Table 1).

**Figure 8 materials-13-05239-f008:**
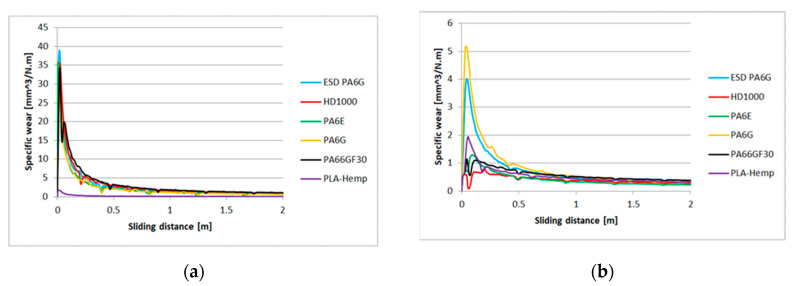
The relation between the specific wear and sliding distance for several polymers types on P150 abrasive clothing: (**a**) lowest applied *pv*, (**b**) highest applied *pv*.

**Figure 9 materials-13-05239-f009:**
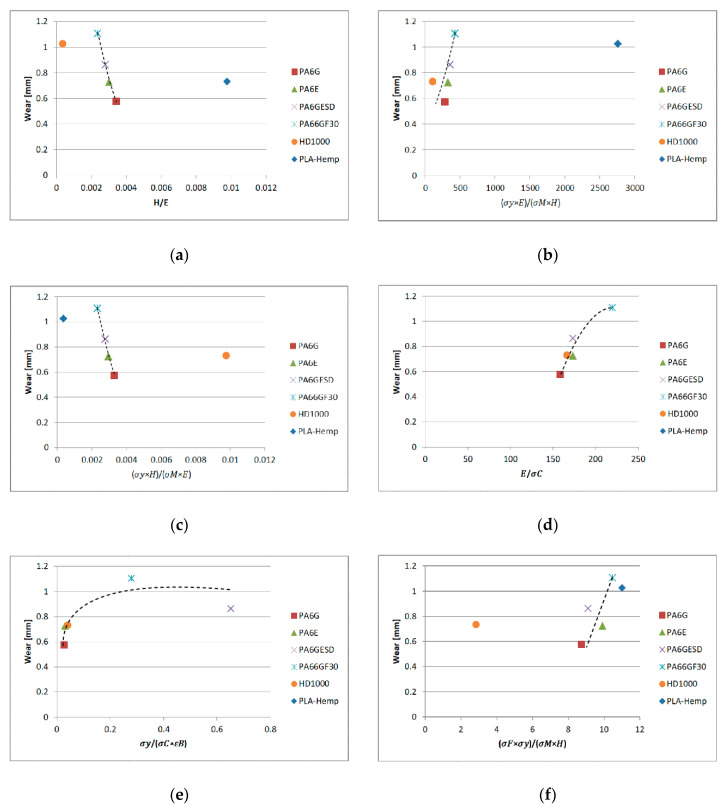
Wear at the end of tests plotted against dimensionless numbers combined with mechanical properties: (**a**) $\frac{H}{E}$; (**b**) $\frac{\sigma_{y}E}{\sigma_{M}H}$; (**c**) $\frac{\sigma_{y}H}{\sigma_{M}E}$; (**d**) $\frac{E}{\sigma_{C}}$; (**e**) $\frac{\sigma_{y}}{\sigma_{C}\varepsilon_{B}}$; (**f**) $\frac{\sigma_{F}\sigma_{y}}{\sigma_{M}H}$.

**Figure 10 materials-13-05239-f010:**
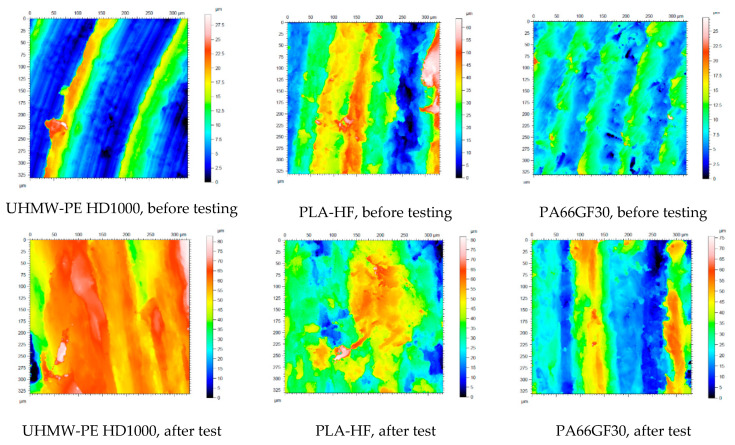
Some microscopic visualization of the change of surface 3D topography of the tested materials on P60 abrasive, applying the top speed and load (No. 6 condition, Table 1).

**Figure 11 materials-13-05239-f011:**
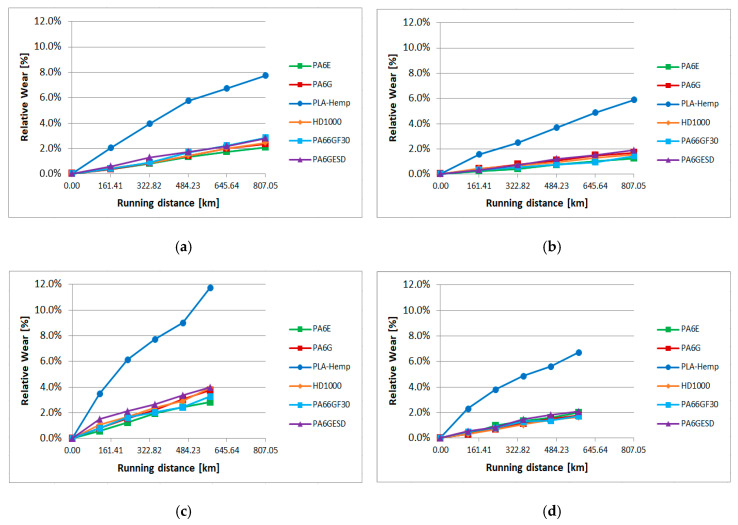
Relative wear of the tested materials in gravelly skeletal soil-based slurry (**a**) position 1, (**b**) position 2, (**c**) position 3, (**d**) position 4 (Figure 3).

**Figure 12 materials-13-05239-f012:**
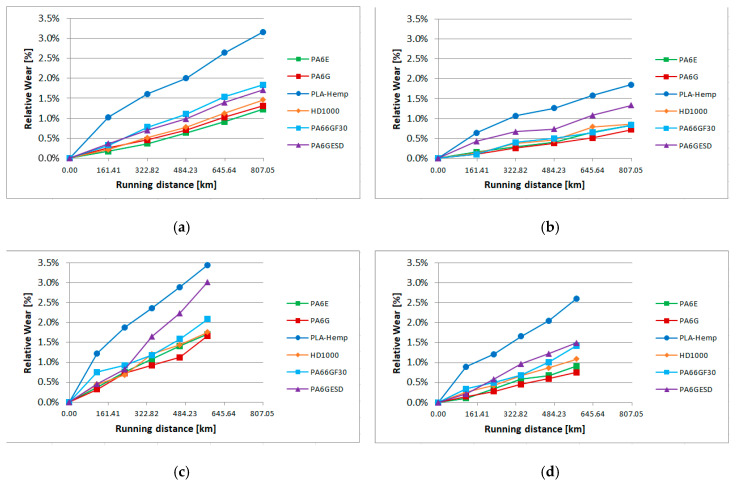
The relative wear of the tested materials in loamy soil-based slurry (**a**) position 1, (**b**) position 2, (**c**) position 3, (**d**) position 4 (Figure 3).

**Figure 13 materials-13-05239-f013:**
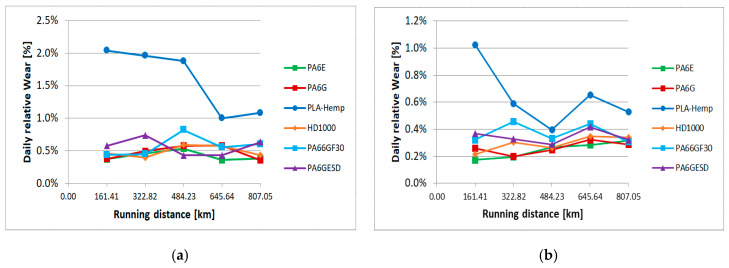
Daily relative wear in position 1 (Figure 3) with skeletal (**a**) and loamy (**b**) soil-based slurry.

**Figure 14 materials-13-05239-f014:**
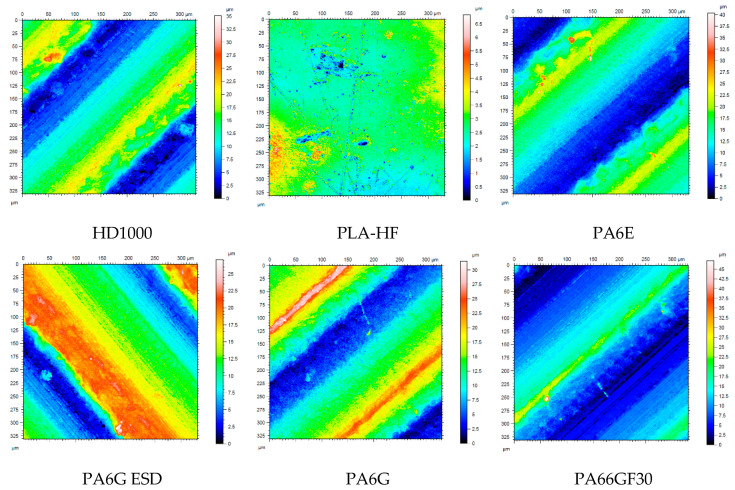
The tested polymer surfaces before testing: PLA-HF is pressed, while the others are machined surfaces.

**Figure 15 materials-13-05239-f015:**
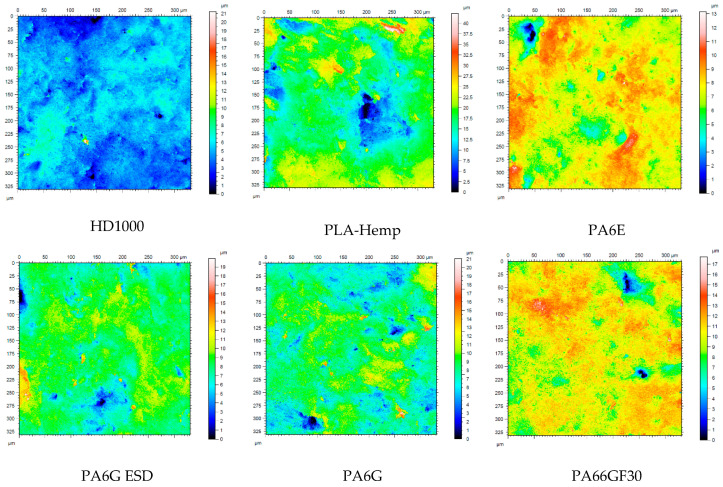
Microscopic visualization of the change of surface 3D topography of the tested materials in gravel slurry at position No.1 (Figure 3).

**Table 1 materials-13-05239-t001:** Features of the twelve test conditions in the abrasive pin-on-plate test system.

Test System No.		1	2	3	4	5	6	7	8	9	10	11	12
Test conditions	Load, *F_N_* (N)	9.81	29.43	49.05	9.81	29.43	49.05	9.81	29.43	49.05	9.81	29.43	49.05
	*v* (m/s)	0.031	0.031	0.031	0.056	0.056	0.056	0.031	0.031	0.031	0.056	0.056	0.056
	Abrasive interface	P60	P60	P60	P60	P60	P60	P150	P150	P150	P150	P150	P150
	Calculated *pv* (MPa·ms^−1^)	0.0062	0.0185	0.0308	0.0109	0.0327	0.0544	0.0062	0.0185	0.0308	0.0109	0.0327	0.0544

**Table 2 materials-13-05239-t002:** Properties of the soils used as abrasive media.

Geotechnical Property	Lime Coated Chernozem (Loamy)	Gravelly Skeletal Soil (Gravel)
Natural moisture content *W* (%)	26.02	12.00
Volumetric weight in the state of natural moisture content *ν* (kN/m³)	18.67	18.96
Specific gravity *νs* (kN/m³)	25.05	26.02
Volumetric weight in the dry state *νd* (kN/m³)	14.80	16.92
The lower limit of plasticity *Wp* (%)	29.58	-
The upper limit of plasticity *WL* (%)	56.00	-
Consistency index *Ic*	1.13	-
Pore index *e*	0.69	0.53
Porosity *n* (%)	40.74	65.60
Edometric deformation modulus *M* (kPa)	9558	12,000
Linear deformation modulus *E* (kPa)	9558	12,000
Internal friction angle *Ф* (°)	13	20
Cohesion *c* (kPa)	100	25
Plasticity criterion *Cp*	26.00	-

**Table 3 materials-13-05239-t003:** Mechanical properties of the used composite materials.

Mechanical Properties	PA6G	PA6E	PA66GF30	UHMW-PE HD1000	PA6G ESD	PLA-HF
Yield stress $\sigma_{y}$ (MPa)	80	78	91	19	75	58.2
Modulus of elasticity (tensile test) *E* (MPa)	3500	3300	5500	750	4000	32,600
Elongation at break $\varepsilon_{B}$ (%)	130	130	13	>50	5	0.6
Flexural strength $\sigma_{F}$ (MPa)	109	100	135	21	102	130.2
Tensile strength $\sigma_{M}$ (MPa)	83	79	91	19	75	58.2
Compressive strength 1% $\sigma_{C}$ (MPa)	22	19	25	4.5	23	0.737
Shore D Hardness (–)	83	75	86	62	80	82.3
Derived Hardness *H* (MPa)	12	9.9	12.9	7.3	11.2	11.8

**Table 4 materials-13-05239-t004:** The slope values (speed of wear) of linear regression in all the 12 test conditions.

Material	1	2	3	4	5	6	7	8	9	10	11	12
PA66GF30	0.1198	0.323	0.5907	0.108	0.3457	0.5676	0.0422	0.0808	0.1053	0.0355	0.0632	0.0838
PA6G ESD	0.0433	0.1296	0.2065	0.0415	0.1036	0.1586	0.032	0.0643	0.0918	0.0255	0.0555	0.0794
PA6E	0.0457	0.1032	0.1439	0.0343	0.0898	0.1184	0.0269	0.0563	0.0757	0.0198	0.0485	0.0648
PA6G	0.0258	0.0549	0.1338	0.016	0.0669	0.082	0.0192	0.0523	0.0753	0.0186	0.048	0.0641
PLA-HF	0.0912	0.1821	0.2165	0.0551	0.1292	0.2151	0.0455	0.0759	0.1036	0.0355	0.056	0.078
HD1000	0.0508	0.171	0.3112	0.0578	0.1598	0.2609	0.0228	0.0228	0.0702	0.0211	0.0428	0.0552

**Table 5 materials-13-05239-t005:** Coefficients of the regression model with P60 abrasive.

Model	Coefficient	Standardized Regression Coefficient, Beta	*t*	*p*
Constant	−0.387		−33	<0.001
s	0.125	0.707	539	<0.001
$F_{N}$	0.501	0.599	457	<0.001
$\frac{\sigma_{y}}{\sigma_{c}\varepsilon_{B}}$	0.010	0.341	239	<0.001
H	−0.066	−0.092	−65	<0.001

**Table 6 materials-13-05239-t006:** Coefficients of the regression model with P150 abrasive.

Model	Coefficient	Standardized Regression Coefficient, Beta	*t*	*p*
Constant	−0.157		5.9	<0.001
s	0.063	0.756	23.5	<0.001
$F_{N}$	0.260	0.598	−4.4	<0.001
v	−5.987	−0.105	−8.8	<0.001
$\frac{\sigma_{y}}{\sigma_{c}\varepsilon_{B}}$	0.003	0.176	−5.1	<0.001

**Table 7 materials-13-05239-t007:** Original surface characteristics of the tested polymers.

	HD1000	PLA−HF	PA6E	PA6G ESD	PA6G	PA66GF30
*Sq* (μm)	5.3	13.1	1.9	1.9	1.5	2.6
*Ssk*	1.0	0.1	0.4	0.2	−0.2	0.5
*Sku*	3.0	2.3	1.9	1.9	2.0	4.1
*Sp* (μm)	21.6	36.0	5.7	4.6	3.4	18.9
*Sv* (μm)	7.6	27.4	3.3	3.7	3.9	8.3
*Sz* (μm)	29.3	63.5	9.1	8.3	7.3	27.2
*Sa* (μm)	4.4	10.8	1.6	1.7	1.3	2.1

**Table 8 materials-13-05239-t008:** Surface characteristics after the tests and the change in %.

	HD1000	PLA-HF	PA6E	PA6G ESD	PA6G	PA66GF30
	after the test					
*Sq* (μm)	9.5	12.6	9.4	12.7	8.2	13.2
*Ssk*	−2.1	0.01	0.2	−0.004	0.3	0.5
*Sku*	12.4	2.8	3.6	2.1	2.8	2.2
*Sp* (μm)	25.9	45.4	27.9	33.5	31.5	49.1
*Sv* (μm)	56.8	36.2	30.6	27.3	18.6	26.5
*Sz* (μm)	82.7	81.6	58.5	60.8	50.1	75.7
*Sa* (μm)	6.3	10.1	7.0	11.1	6.7	11.1
	Change in %					
*Sq* (μm)	104%	14%	672%	864%	709%	149%
*Ssk*	−134%	−214%	−169%	−155%	401%	4%
*Sku*	−19%	27%	51%	57%	141%	−9%
*Sp* (μm)	51%	6%	495%	996%	1131%	57%
*Sv* (μm)	285%	49%	1142%	1477%	948%	116%
*Sz* (μm)	112%	24%	733%	1212%	1033%	75%
*Sa* (μm)	109%	7%	621%	786%	606%	138%

**Table 9 materials-13-05239-t009:** Ranking the abrasive sensitivity to the system features.

Factors in Increasing Abrasive Sensitivity to System Variables
	Less Dominant	More Dominant
	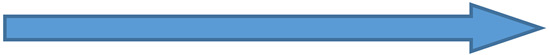
Wear, P60	H	$\frac{\sigma_{y}}{\sigma_{c}\varepsilon_{B}}$		
Wear, P150	$\frac{\sigma_{y}}{\sigma_{c}\varepsilon_{B}}$			
$F_{f}$, P60	$\varepsilon_{B}$	$\frac{\sigma_{c}}{\sigma_{y}\varepsilon_{B}}$		
$F_{f}$, P150	H	$\frac{\sigma_{F}\sigma_{y}}{\sigma_{M}H}$		
ΔT, P60	H	$\frac{H\varepsilon_{B}}{\sigma_{y}}$	$\frac{\sigma_{F}H}{\sigma_{M}E}$	$\sigma_{F}$
ΔT, P150	$\varepsilon_{B}$	H	$\frac{\sigma_{F}H}{\sigma_{M}E}$	
*Sq*, P150	E			
*Ssk*, P60	$\sigma_{F}$			
*Ssk*, P150	H			
*Skv*, P150	E			
*Sp*, P150	$\frac{\sigma_{y}}{\sigma_{c}\varepsilon_{B}}$			
*Sv*, P150	E			
*Sz*, P150	$\frac{\sigma_{F}}{\sigma_{c}}$			
*Sa*, P150	E			

**Table 10 materials-13-05239-t010:** 3D surface characteristics before testing.

	HD1000	PLA-HF	PA6E	PA6G ESD	PA6G	PA66GF30
*Sq* (μm)	5.5	0.6	5.9	5.6	6.1	5.3
*Ssk*	0.02	0.5	0.04	−0.1	0.2	0.6
*Sku*	2.0	5.6	2.2	1.9	2.5	3.5
*Sp* (μm)	23.6	4.1	28.5	14.5	19.7	37.3
*Sv* (μm)	11.4	2.7	11.7	12.5	11.7	9.7
*Sz* (μm)	35.09	6.8	40.3	27.07	31.5	47.1
*Sa* (μm)	4.6	0.4	5.0	4.7	5.0	4.3

**Table 11 materials-13-05239-t011:** Surface characteristics after the tests and the change in %.

	HD1000	PLA-HF	PA6E	PA6G ESD	PA6G	PA66GF30
	after the test					
*Sq* (μm)	1.2	4.5	1.2	1.5	1.9	1.7
*Ssk*	0.4	−0.2	−0.9	−0.4	0.02	−1.02
*Sku*	11.3	4.5	6.5	6.1	4.4	6.8
*Sp* (μm)	16.3	25.0	5.5	11.7	12.9	7.5
*Sv* (μm)	4.8	17.3	7.6	8.1	8.0	10.1
*Sz* (μm)	21.2	42.3	13.1	19.9	21.0	17.7
*Sa* (μm)	0.9	3.4	0.9	1.1	1.52	1.2
	Change in %					
*Sq* (μm)	−77.57%	622.75%	−78.95%	−71.89%	−67.73%	−67.55%
*Ssk*	1796.28%	−148.57%	−2088.16%	237.37%	−91.08%	−255.53%
*Sku*	448.19%	−20.51%	194.21%	213.85%	74.80%	91.78%
*Sp* (μm)	−30.72%	507.86%	−80.59%	−19.61%	−34.42%	−79.70%
*Sv* (μm)	−57.72%	537.67%	−35.48%	−34.48%	−31.49%	3.87%
*Sz* (μm)	−39.54%	519.71%	−67.43%	−26.48%	−33.32%	−62.43%
*Sa* (μm)	−79.88%	659.34%	−81.64%	−75.61%	−69.69%	−70.67%

**Table 12 materials-13-05239-t012:** Ranking the abrasive sensitivity to slurry system features.

Factors in Increasing Abrasive Sensitivity to System Variables
Less Dominant More Dominant
	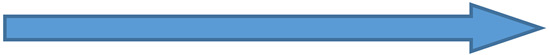
Wear, gravel	$\frac{\sigma_{y}}{\sigma_{c}\varepsilon_{B}}$		
Wear, loamy soil	$\sigma_{F}$	$\varepsilon_{B}$	$\frac{\sigma_{y}E}{\sigma_{M}H}$
*Sq*, loamy soil	E		
*Ssk*, gravel	$\frac{\sigma_{y}E}{\sigma_{M}H}$	E	
*Sa*, loamy soil	E

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
