# Peer review of "Abrasive Sensitivity of Engineering Polymers and a Bio-Composite under Different Abrasive Conditions"

_materials, 2020, doi:10.3390/ma13225239_

Round 1
Reviewer 1 Report
The research appears to be competently conducted with experiments and statistical analysis of the results. The research illustrates the sensitivity of different materials and material properties suitable for harvesting machinery.
The English needs careful correction. It would help if this was completed by a native speaker.
Some of the colours on the figures are difficult to distinguish. Could these be made clearer in some way.
The F values are given to ridiculous and laughable accuracy. For example, 10 significant figures! 3 or at most 4 significant figures is appropriate.
Author Response
Dear Reviewer,
I attached the answers in file.

Reviewer 2 Report
The authors show the abrasive sensitivity of plastics. It is important to research materials of mechanical parts. However, the manuscript should be revised before publication.
The characters indicating physical quantity should be italic or oblique style. For example, inline 128, page 3, “v=2.4 m/s” is shown. This “v” should be changed into an italic or oblique style.
Labels of microscopic visualization shown in Figs 10, 14, and 15 are too small. The authors should enlarge such labels.
The tables and figures should not be divided into 2 pages. For example, Table 4 is divided into pages 10 and 11.
In the case of pin-on-plate experiments, bio-degradable composite performed an averaged wear resistance. However, the relative wear of PLA-Hemp is very high. What is the reason for this difference? The readers of this journal want to know the scientific explanation for this difference. The reviewer thinks that this viewpoint is the most important in this study.
Author Response

(The authors gave the same response as above.)

Reviewer 3 Report
The authors have carried out extensive tribological research, which shows many interesting observations. Nevertheless, the multitude of studies and the desire to present all the most interesting results meant that they were not described well and precisely. There is a lack of information on the sequence of actions, especially calculations, and there are also some important test results, e.g. in the context of the temperature measured during the friction process. I understand that it was the authors' intention to present the maximum amount of information in one article without extending it too much. It didn't quite work out. The article, although relatively long, still has serious substantive gaps. I recommend rebuilding it or splitting it into two. Without loss of knowledge, it is possible to separately describe the results of research on one and then the other research method, because there was no direct translation between the first and the other results in the work. Regardless of the authors' decision, I recommend paying attention to the issues presented below.
Line 48
In the context of the information on environmental protection, please bear in mind that the use of polymeric materials in friction junctions in agricultural machinery is not beneficial for the environment by itself due to the micro-scraps of material leaking into the soil. In addition, farmers very often do not care for used machine parts and they are very often found on farms. Maybe it's worth changing this opinion a bit?
Lines 69-90
This type of information is more relevant to review articles. There is no need to describe the types of wear and their mechanisms in the research article, unless the article concerns the evaluation of the mechanism itself.
Line 203 (e.g)
It is necessary to explain what is pv at the first appearance of this symbol. The same remark applies to all other symbols.
Line 240
Was the temperature controlled in the pin-on-plate test as well?
Line 244
It should be clarified what was dictated by the duration of both tests. Due to the diametrically different times, it is difficult to understand the motivation for selecting these tests in parallel to evaluate material properties.
Line 275
Where do the data in table 3 come from? Did the authors perform the appropriate experimental tests themselves? The information should be specified, the more so as it seems to be the basis for further calculations.
Line 312
How has it been verified that linear regression will be most adequate?
Line 316
The F-test assumes that the test results are normally distributed. Has it been verified?
Line 340
Where are that linear regression?
Line 348
This wording is unclear. have the significance tests been performed? Are the data in Table 3 average of the results? How many?
Line 350
Do the test results shown in Figures 7 and 8 correspond to any of the specific settings shown in Figure 3?
Line 373
This paragraph indicates that in addition to abrasive wear, other test results were analyzed. The article does not present them, however, without indicating their physical meaning, mathematical models, e.g. concerning temperature, were presented at once (6).
Model (2) was described as the best. It does not follow from the description above in the article why the mathematical model does not include the associated coefficient, eg E/σC. How do you know that e.g. the H/E ratio has no significant effect on consumption? And that assumption apparently worked.
Moreover, it is not understood where the variables "s" and "FN" came from in the model, since it was not mentioned earlier that they would be included? The F test was performed only for the coefficients presented in the model (2) and (3). The test shows that the coefficients from the model are at the same level of significance, which additionally raises the aforementioned doubts.
Similar questions arise when reading the following mathematical models.
Line 463
It is not clear how the sensitivity ranking of the coefficients in Table 9 was developed. Table 9 is not referenced in the text.
Line 371
Two methods of conducting the friction process were used in the study, information about one of them is dominant in conclusions.
Author Response

(The authors gave the same response as above.)

Round 2
Reviewer 3 Report
The authors introduced a few clarifications and the article became more understandable. Nevertheless, a few more notes remain to be considered (numbers of lines refer to notes on the first manuscript and the responses to them).
Line 203
Unfortunately, still other symbols appearing in tables and graphs are not explained in the text.
Line 275
This should be clarified in the article.
Line 312
Please check the bibliography, in the "Materials" journal itself you will certainly find the results where the nature of the abrasive wear visually resembles the line of a second-order polynomial.
Line 316
The use of „the simplest approximation of all” model may lead to errors in the interpretation of the results. Has this been taken into account?
Line 340
Please also modify the title of the table.
Line 348
Based on your explanation please specify the text of the article.
Line 371
Please indicate at least the mutual relations in the results of both methods. The last sentence is too general and does not summarize the validity of using both types of research. It should be more clearly stated how the results from the pin on plate tests will apply to slurry-pot applications, in particular, to compare the results of these tests. And I miss straightforward written conclusions related to the application of mathematical models in practice.
